# Multiverse Predictions for Habitability: Element Abundances

**McCullen Sandora** [1,*]**, Vladimir Airapetian** [2,3]**, Luke Barnes** [4]**, Geraint F. Lewis** [5] **and Ileana Pérez-Rodríguez** [6]

1 Blue Marble Space Institute of Science, Seattle, WA 98154, USA
2 Sellers Exoplanetary Environments Collaboration, NASA Goddard Space Flight Center, Greenbelt, MD 20771, USA
3 Department of Physics, American University, Washington, DC 20016, USA
4 School of Science, Western Sydney University, Locked Bag 1797, Penrith South Dc, NSW 2751, Australia
5 Sydney Institute for Astronomy, School of Physics, A28, The University of Sydney, NSW 2006, Australia
6 Department of Earth and Environmental Science, University of Pennsylvania, Philadelphia, PA 19104, USA
* Correspondence: mccullen@bmsis.org

**Abstract:** We investigate the dependence of elemental abundances on physical constants, and the implications this has for the distribution of complex life for various proposed habitability criteria. We consider three main sources of abundance variation: differing supernova rates, alpha burning in massive stars, and isotopic stability, and how each affects the metal-to-rock ratio and the abundances of carbon, oxygen, nitrogen, phosphorus, sulfur, silicon, magnesium, and iron. Our analysis leads to several predictions for which habitability criteria are correct by determining which ones make our observations of the physical constants, as well as a few other observed features of our universe, most likely. Our results indicate that carbon-rich or carbon-poor planets are uninhabitable, slightly magnesium-rich planets are habitable, and life does not depend on nitrogen abundance too sensitively. We also find suggestive but inconclusive evidence that metal-rich planets and phosphorus-poor planets are habitable. These predictions can then be checked by probing regions of our universe that closely resemble normal environments in other universes. If any of these predictions are found to be wrong, the multiverse scenario would predict that the majority of observers are born in universes differing substantially from ours, and so can be ruled out, to varying degrees of statistical significance.

**Keywords:** multiverse; habitability; element abundances





## 1. Introduction

There is seemingly an element of arbitrariness to the laws of physics. This has prompted some to speculate that the laws we observe may not be unique, but instead may vary from place to place [1]; this has become known as the multiverse hypothesis [2]. If this is correct, we will never be able to explain our physical laws in the same way we explain other features of the universe, such as the size of an atom, through relating them to more fundamental quantities [3]. In this scenario, many of the universe's peculiar features and behaviors are instead understood by an alternative type of explanation known as the anthropic principle [4,5]. By invoking this, we start with the tautology that we can only exist in universes capable of supporting complex life, and use ensuing selection effects to explain otherwise puzzling aspects of our universe. For example, it was put forward in [6] that our universe is as immense as it is, even though the theoretical expectation for typical universe size is minuscule, because a universe must be practically as large as ours to host galaxies large enough to retain the heavy elements necessary for planet formation. A major criticism of these anthropic arguments is that we do not know the conditions complex life requires to arise and thrive, since our current understanding of life is based solely on the single example of Earth life, which shares a common ancestor. Consequently, invoking anthropics to explain any fact about our whereabouts, its critics would have, makes strong assumptions about the nature of habitability. This leaves us vulnerable to overlooking a more fundamental, reductionistic explanation [7], akin to how early anthropic explanations for the length of the year overlooked the fact that this can be derived through Kepler's

third law. Because it is unlikely that we will ever have the opportunity to directly verify the existence of other universes, anthropic explanations run the risk of not meeting the basic criteria that define scientific practices, and invite a potentially unending debate [8].

This situation can be remedied by inverting this anthropic logic; instead of committing to a set of assumptions about habitability and inventing post-hoc explanations for our observations, we may consider many potential habitability criteria and determine which are compatible with the fact that we are in this universe. This reasoning uncovers preferences for particular habitability criteria over others, generating predictions for which conditions are expected to be necessary for complex life. Here, compatibility with observations is judged by characterizing how the features of universes (such as, for example, the lifetime of stars and the size of planets) vary, allowing us to determine which properties of our universe are generic throughout the multiverse, and which are exceptional. By calculating the probability distributions of observing these features within the multiverse, we can compare different habitability conditions on the basis of how likely each makes our location here. If some particular assumptions about habitability indicate that the majority of complex life is in other universes with different properties, these assumptions would imply that our presence in this universe is exceedingly atypical. By invoking the principle of mediocrity [9], which states that we expect to be typical among all observers who make a particular observation, we can declare that conditions making our observations unlikely are incompatible with multiverse expectations. In turn, this generates predictions for which conditions are necessary for habitability. In the coming decades, we will have the opportunity to learn much more about the nature of habitability, and will be able to probe regions of our universe that closely resemble normal environments in universes that are predicted to be uninhabitable. If our findings run counter to these predictions, this will serve to falsify the multiverse hypothesis, often to a very high degree of confidence.

This task requires us to be able to quantitatively assess the probability of being in a particular universe. Among other things to be discussed below, the probability of being in a particular universe is proportional to the number of observers that universe contains. The term 'observer' usually means something like 'a conscious entity' but is left ambiguous here, as to date it has no consensus definition. We may brush aside philosophical discussions about who exactly qualifies because we make the simplifying assumption that observers roughly correlate with complex life, here defined as multicellular organisms (this definition may in fact reflect untoward bias; see [10]). In addition, we make an even greater assumption for the present paper, which is that the conditions necessary for complex life do not differ substantially for those necessary for simple life (unicellular organisms). To be sure, unicellular organisms can survive over a broader range of conditions than multicellular life, but, when compared to the full range of possible environmental variability, the differences no longer appear so significant. Both of these assumptions are meant to be approximations aimed at facilitating our calculations, and have been partially examined in previous papers of this series [11,12]. However, further work is needed to determine what interesting conclusions arise from considering the differences in distributions between simple life, complex life, and observers. These assumptions allow us to refer to the habitability of a universe as the number of observers it contains. Via our assumptions, the habitability of a universe is then also equated with the amount of simple life it can support.

Our strategy for evaluating different habitability conditions can be employed in a rather formulaic manner toward any that one deems important. Life's occurrence depends on a great many different conditions, each of which inhibit or promote habitability to different degrees when parameters are changed. The present paper is part of a series that sets to task programmatically incorporating as many habitability conditions into this framework as possible. To begin, we focused on how the number and properties of stars vary with physical constants (taken to mean masses of particles and strengths of forces throughout) in [13], followed by the occurrence and properties of planets [14], the fraction of planets that develop life [11], and the rates of mass extinctions [12]. This exercise has so far yielded around a dozen predictions for various habitability criteria, any of which would be capable of ruling out the multiverse in the range of $2 - 6\,\sigma$ (with probability ranging

from 0.05 to $10^{-10}$) if found false. However, our analysis is by no means complete, and sustained effort is required to thoroughly fulfill the potential of this enterprise.

Any reliance of life on the particular chemistry present in an environment was neglected in our previous works, enforcing a de facto stance that habitability of an environment should be completely insensitive to its chemical composition. Here, we investigate the effects of adopting various expectations for the role of chemical abundances and life. For this, we consider three different effects that can drastically alter elemental abundances, and their consequences on purportedly essential elements: variations in the metal-to-rock ratio, alpha burning in massive stars, and isotope stability.

The first is the ratio of metals to rocks, which is set by the relative yields of the supernovae which produce each type of element. In Section 2, we find that an upper bound on this ratio, proposed to maintain the stability of liquid surface water, is disfavored by, though only mildly incompatible with, the multiverse, and that any lower bound bears no consequence on the probabilities of our observations.

In Section 3, we investigate the dependence of alpha burning in massive stars on the Hoyle resonance, and the implications this has on the carbon-to-oxygen ratio, the magnesium-to-silicon ratio, and the organic-to-rock ratio (defined as (C+O)/(Mg+Si), with no intent to imply that all carbon and oxygen are contained in biomolecules, or discount the importance of hydrogen). We find, in line with some of the original anthropic arguments, that the organic-to-rock ratio must be important for life to be compatible with the multiverse. Further, a planetary threshold for the magnesium-to-silicon ratio which is quite close to Earth's value must be unimportant.

Lastly, in Section 4 we consider the variations in isotope stability with the physical constants. We find that this induces large and drastic changes to the abundances of every element throughout parameter space, and consider the effects this has on nitrogen, phosphorus, sulfur, and iron. The expectations that nitrogen and phosphorus are crucial for habitability are disfavored, though only mildly incompatible with the multiverse, and no predictions can be made for sulfur and iron. Including the carbon-to-oxygen criterion even alleviates the nitrogen and phosphorus incompatibilities with the multiverse. Other elements essential for Earth biochemistry, such as hydrogen, boron, sodium, and chlorine, do not exert appreciable influence on our calculations, and so are not considered in depth.

In our analysis, we make use of previously described criteria that were determined to give the best probabilities for observing our values of the physical constants. These are: the yellow condition, which enforces that stellar radiation is within a narrow window of atomic energies (taken to be 600–750 nm), so that photosynthesis is possible [13], and the entropy condition, which treats the probability of the emergence of life on a planet as directly proportional to the incident entropy, in the form of stellar radiation [11]. Note that the yellow condition is equivalent to restricting attention to stars with long lived, large habitable zones [15] and rarefied, cooler winds [16], which are almost necessarily corollaries of stellar surface temperature being similar to molecular binding energies. These are not the unique criteria so far uncovered that would make all observations that we consider viable, and we discuss alternative possibilities more thoroughly in Section 5.

We also assume that a planet must be temperate (so that surface liquid water is stable) and terrestrial (to retain heavy but not light gases) in order to qualify as habitable. These restrictions are based on the expectation that liquid water is necessary for life, in accordance with [17,18]. Indeed, the presence of water is observed to place rather strict limits for life, with very few microbes capable of being biologically active under extreme desiccation [19,20]. Our discussion presupposes both that water retains its properties which are deemed essential for habitability as the physical constants are varied, that water will be present on planets where it has the potential to be liquid, and, for some planets at least, in moderate enough abundance to maintain some land. These may not necessarily be true, but are not investigated here. Ignoring these effects is tantamount to assuming that these properties are not essential, assumptions which will be investigated in more detail in future work. The list of habitability assumptions we employ, along with the conditions we investigate in this work, is displayed in Table 1.

**Table 1.** List of habitability conditions assumed/examined in this work. Each condition is assumed necessary for habitability, with justification/analysis provided in the specified locations. Conditions not explicitly mentioned, such as the importance of tidal locking or the properties of water, are by default assumed not to be relevant for habitability in this work, but many have been/will be examined in other papers of this series.

| Assumption | Discussion |
| --- | --- |
| photosynthesis | [13] |
| temperate planets | [14] |
| terrestrial planets | [14] |
| life $\propto$ entropy | [11] |
| observers $\approx$ complex life | [12] |
| complex life $\approx$ simple life | [11] |
| water | future work |
| metal/rock | Section 2 |
| C, O, Mg, Si | Section 3 |
| N, P, S, Fe (Na, Cl) | Section 4 |

Our previous analyses were restricted in scope to variations of only three of the physical constants of nature: the fine structure constant $\alpha = e^2/(4\pi)$, the electron-to-proton mass ratio, $\beta = m_e/m_p$, and the strength of gravity $\gamma = m_p/M_{pl}$, where $e$ is electric charge, $m_e$ is the electron mass, $m_p$ the proton mass, and $M_{pl}$ the Planck mass. These sufficed for the previous macroscopic properties we considered. However, since the elemental abundances also depend sensitively on the quark masses, we take this opportunity to enlarge the parameters which we vary to include the up and down quark masses, $m_u$ and $m_d$. The additional quantities we use to parameterize these are denoted by $\delta_u = m_u/m_p$ and $\delta_d = m_d/m_p$. Discussion of how these are implemented into our existing computations are given in Section 6.

Evaluating a given habitability hypothesis involves determining the probability of observing our universe's values of each of these constants, computed according to the formula $p(\vec{x}) \propto N_{obs}(\vec{x}) p_{prior}(\vec{x})$, where $\vec{x}$ are the constants, $N_{obs}$ is the number of observers (conscious life forms) in a given universe, and $p_{prior}$ is the underlying distribution of universes[1]. As discussed in [13], we take as a reasonable measure that $p_{prior} \propto 1/(\beta\gamma\delta_u\delta_d)$, though most of our results do not depend on this precise choice too much[2]. Far more important is the quantity $N_{obs}$, which depends sensitively on the assumed requirements for complex life. The probability of observing our measured value of any constant is defined as $\mathbb{P}(x_{obs}) = \min(P(x < x_{obs}), P(x > x_{obs}))$. For reference, the probabilities of observing the five values of our constants with the yellow and entropy baseline assumptions are

$$\mathbb{P}(\alpha_{obs}) = 0.423, \ \mathbb{P}(\beta_{obs}) = 0.273, \ \mathbb{P}(\gamma_{obs}) = 0.234, \ \mathbb{P}(\delta_{u\,obs}) = 0.167, \ \mathbb{P}(\delta_{d\,obs}) = 0.494 \quad (1)$$

These will serve as the basis of comparison for all other habitability criteria we consider in this paper.

## 2. What Metal-to-Rock Ratio Is Required for Life?

Though life is assembled primarily out of dissolved rocks, metals play a crucial role as well. In many environments, they can serve as the limiting nutrient, setting population sizes [22]. Metals are also implicated in leading origin of life scenarios [23,24], and have been argued to be essential for the origin of many primordial anabolic processes [25] given that they serve as cofactors catalysing many important biochemical processes. Conversely, the high reactivity of metals makes them toxic when present in large quantities [26]. Indeed, regions of high metal pollution are typically restricted to microorganisms adapted to such environments [27]. Therefore, from the biological context, it is not unreasonable to think that there are tolerability limits to the amount of metals, outside of which life would be extremely indigent, if not altogether absent. However, the optimist may disregard such

hesitations, pointing out that several other essential molecules can be just as deleterious, including both free oxygen [28] and water [29]. The precise habitability limits on metals are therefore open to debate at the moment, especially when considering the diversity of single-celled and multicellular lifeforms- as well as what it means to be alive, dead or dormant. Since the metal-to-rock ratio varies throughout the multiverse, we can determine which limits are compatible with human existence in this universe, and use this to make predictions for what to expect these limits to be.

Aside from the biochemical aspects, metals have a significant effect on the planetary system as a whole. These can be examined by considering the metal-to-rock ratio *R*, where we take the word 'metal' here to mean the first row of transition metals in the periodic table, and the word 'rock' to be all other light elements above atomic number 5. The primary effect the metal-to-rock ratio has on a planet is in the determination of core size. Several studies have addressed the effects an altered core size would have on an Earth-like planet: [30] investigate the influence of core size on atmospheric outgassing, and found insufficient outgassing for maintaining surface liquid water in planetary bodies where *R* is greater than 1.9 times Earth's value. They also stress that if the Earth's metal-to-rock ratio were substantially different, plate tectonics would be disrupted, leading to a potentially uninhabitable planet [31]; if *R* is increased to eight times its value, cooling of the much thinner convective region becomes more efficient, and plate tectonics quickly shuts down. If *R* is decreased to 0.08, heat dissipation is very slow and too much atmospheric $CO_2$ builds up. These conclusions are corroborated in the recent analysis done in [32][3].

In principle, extreme metal paucity would adversely affect planetary habitability as well, most notably since the liquid core is responsible for maintaining Earth's magnetic field, buffering the atmosphere from cosmic rays. However, this threshold is incredibly low: since a planet's magnetosphere is linearly proportional to core radius [34], the Earth's would be just as effective at shielding the atmosphere against solar wind above $R = 0.001$. Lastly, a lack of metals will have an adverse effect on stellar dynamos, which act as engines for stellar wind while preventing the erosion of thick planetary atmospheres and non-equilibrium prebiotic chemistry (see [35] for a recent review). However, this threshold is similarly quite weak. Though at this time we cannot be certain which of these thresholds guarantee the absence of complex life, we are in a position to see which are compatible with the multiverse.

Heavy elements are created in stellar fusion, and distributed throughout the surroundings during supernovae events. Of note is the diversity of stellar fates, as the properties of a supernova depend on stellar mass, spin, metallicity, and whether a nearby companion is present. Several types of supernovae are particularly relevant for our purposes: rock-like elements (typified by oxygen and carbon) are produced in type II supernovae, which result when a massive star exhausts its nuclear fuel. A related source of rock-like elements is in the stellar winds and explosions of intermediate stars, which have reached the asymptotic giant branch (or Wolf-Rayet) stage of their evolution. Type Ia supernova arise when a white dwarf siphons enough gas from a nearby companion to exceed the critical Chandrasekhar mass, triggering a collapse and producing most of the metals present in our universe. Because these two processes have such different origins, their yields will depend on physical constants differently, and the ratio between these two processes will vary in other universes. Additionally, type Ia supernovae are typically much more delayed, causing the ratio to change with time. Using a simple star formation model, we compute the distribution of values for a given set of constants. We then integrate this criterion into our existing framework for multiverse probability calculations for several proposed threshold values. To do this, we first consider the two production rates in turn.

The type II production rate is comparatively simple: the rock production rate is given by:

$$\dot{M}_{\text{rock}}(t) = \psi_{\text{SFR}}(t - t_{\text{II}}) \, f_{\text{II}} \, f_{\text{ej}} \, \langle M \rangle_{\text{II}} \qquad (2)$$

Here, $\psi_{\text{SFR}}$ is the star formation rate, $f_{\text{II}}$ is the fraction of stars that are large enough to undergo complete nucleosynthesis up to nickel and iron, $f_{\text{ej}}$ is the fraction of the star's

mass which is ejected as heavy elements during the supernova, and $\langle M \rangle_{\rm II}$ is the average mass of a massive star. The star formation rate is evaluated at a slightly delayed time $t - t_{\rm II}$ to reflect the typical lifetime of these massive stars, but in practice this matters little because the lifetime of massive stars is orders of magnitude smaller than the star formation timescale for practically all parameter values.

The fraction of stars which undergo type II supernovae explosions is about 0.01 and is, to first approximation, independent of physical constants. Briefly, the reason is that this mass scale is dictated by whether the stellar interior will have enough pressure to overcome a nuclear burning threshold [14]. Since both the minimum stellar mass and the knee of the stellar initial mass function also only depend on a similar criterion (albeit for hydrogen burning rather than oxygen), the fraction of massive stars is independent of the details which set these thresholds, and thus does not depend on physical constants.

Similarly, the fraction of mass ejected as heavy elements $f_{\rm ej}$ is roughly independent of the values of the physical constants that we are considering. As described originally in [4], the ejection mechanism relies on the fact that the neutrino outflow during the final stages of the burning process interacts with the outer material, triggering a blowout. This, together with changes in the weak scale, would make the neutrino interaction too weak or too strong, and the material ejected would be significantly diminished. Enforcing that type II supernovae are operational effectively imposes a relatively narrow range of values the weak scale (Higgs vacuum expectation value) may take, though fully incorporating a varying weak scale into our analysis is left for future work. The ejected mass fraction is independent of the constants, essentially because a majority of nuclei within the star must participate in order to trigger a supernova. This scaling was verified by computations performed in [36]. Equally important to note is that the type of material ejected is also independent of constants: the fact that the ejecta is primarily oxygen, even though heavier elements are produced in the star's center, hinges on the fact that the nuclear burning timescale during the final stages of the supernova are much shorter than the Kelvin-Helmholtz turnover timescale, so that the outer material does not have a chance to completely fuse. The fact that the inner material is not ejected is similarly robust, as the radius at which material escapes is always larger than the core [37]. The rock production rate, after all these considerations, is then $\dot{M}_{\rm rock} \sim \psi_{\rm SFR} \alpha^{3/2} \beta^{-3/4} M_{pl}^3 / m_p^2$.

We briefly consider elemental enrichment due to intermediate mass stars: these yield comparable amounts as type II supernovae, though which process dominates depends on environmental conditions and element [38]. Depending on the element, the majority of production in intermediate mass stars can take place from stellar winds [39]. However, like in type II supernovae, the total yield is an O(1) fraction of the star's initial mass, since each element is either ejected through wind until depleted or released in the final supernova. Because of this, the yield from this source scales in the same way as our estimates for rock production above, and so can be treated as a rescaling of our previous estimate. It is also worth mentioning that intermediate mass stars contribute subdominantly to the production of many metals. Therefore, as long as this process is operational, it may prevent universes from being completely metal free. However, as we will see, the lower bound on $R$ is not constraining, and so neglecting this source will not alter our conclusions.

Modeling the type Ia rate is more involved, as the physics dictating this process is more complex. We can express it as

$$\dot{M}_{\rm metal}(t) = \int_0^t dt_{\rm I}\, \psi_{\rm SFR}(t - t_{\rm I})\, f_{\rm WD}\, f_{\rm binary}(a_{\rm Roche})\, f_{\rm IMF}(t_{\rm I}) \langle M \rangle_{\rm WD} \tag{3}$$

There are several important factors here: $f_{\rm WD}$ is the fraction of stars which are large enough to undergo a supernova explosion within the lifetime of the system, but small enough that the remnant is a white dwarf rather than a black hole or neutron star[4]. Secondly, $f_{\rm binary}(a_{\rm Roche})$ is the fraction of stars which occur in binaries, which are sufficiently closely spaced for Roche lobe overflow to trigger catastrophic mass transfer, resulting in explosion. Most importantly, $f_{\rm IMF}(t_{\rm I})$ is the fraction of companions which deplete their hydrogen

within time $t_I$. This results a tenfold expansion in stellar radius (independent of constants), such that nearly all type Ia supernovae that occur are for companions which have entered the helium burning phase [41]. Lastly, $\langle M \rangle_{WD}$ is the average white dwarf mass, which, like the average large star mass, is dimensionally set by the Chandrasekhar mass, $M_{Ch} \sim M_{Pl}^3 / m_p^2$. Here we must integrate over the lifetime of these systems, as the spectrum of stellar lifetimes is quite broad, and ranges over the galactic depletion timescales.

Our treatment of the fraction of binary stars $f_{binary}(a_{Roche})$ will be cursory and superficial here, with the intent to convince the reader that this does not depend much on the constants we vary. A more detailed account is given in the Appendix A. However, it suffices to note that the distribution of initial separations is log-uniform [42]. As such, the fraction closer than any given threshold is only logarithmically sensitive to the parameters in question. We find that, over the entire range of parameters we vary, this quantity only changes by several percent.

Turning now to $f_{IMF}(t_I)$, we use the simple Salpeter power law form for the initial mass function, $f_{IMF}(\lambda) = (\beta_{IMF} - 1)\lambda_{min}^{\beta_{IMF}-1}/\lambda^{\beta_{IMF}}$, where $\lambda$ is stellar mass, in units of the Chandrashekar mass, $\lambda_{min}$ is the smallest stellar mass, and $\beta_{IMF} = 2.35$ [43]. This neglects two things: firstly, the power law form used here is not valid for small stellar masses. This is unimportant for the regimes we are interested in, as those stars will not start helium burning early enough to be relevant, and so will only affect a shift in the normalization of this distribution. Secondly, it treats binary companion masses as uncorrelated. This assumption does not hold for close binaries [44], where it is found that the mass ratio tends to 1. However, use of this distribution is justified, as the companion star can be treated as drawn randomly from the initial mass function, and so marginalizing over this reproduces the original distribution, regardless of any correlation.

To turn this into a distribution for stellar lifetimes, we use the following expression for dependence of lifetime on mass: $t_\star(\lambda) = 110\alpha^2 M_{pl}^2/(\lambda^{5/2}m_e^2 m_p)$ [13]. Lastly, we use the following expression for the star formation rate: $\psi_{SFR}(t) = \psi_0 \exp(-t/t_{dep})$, where the depletion time is simply related to the galactic freefall time, so that $t_{dep} \sim (G\rho)^{-1/2} \sim 0.070 M_{pl}/(\kappa^{3/2}m_p^2)$ [14], normalized to 3.6 Gyr in our universe from [45]. The density parameter $\kappa = Q(\eta\omega)^{4/3}$, with $Q$ the amplitude of primordial fluctuations, $\eta$ the baryon-to-photon ratio, and $\omega$ the matter-to-dark matter ratio. This star formation prescription is perhaps somewhat simplistic, but it suffices to indicate how strongly the metal-to-rock ratio varies. We plan on returning to more realistic models of star formation, taking galactic substructure and superstructure into account, in future work.

The production rate then becomes

$$\dot{M}_{metal}(t) = f_{binary}\langle M \rangle_{WD}\, \psi_0\, e^{-t/t_{dep}} \times \int_0^{t/t_{dep}} dz\, e^z\, \frac{d}{dz}\left(\frac{z}{z_{max}}\right)^{\zeta_{IMF}} \tag{4}$$

Here $z_{max} = t_\star(\lambda_{min})/t_{dep} = 62100\, \kappa^{3/2}/(\alpha^{7/4}\beta^{1/8}\gamma)$ and $\zeta_{IMF} = 2/5(\beta_{IMF} - 1) = 0.54$. Then the metal-to-rock ratio of stars produced at time $t$ (normalized to equal 1 for solar composition for ease of exposition) becomes

$$R(t) = 7311\, \alpha^{-3/2}\, \beta^{3/4}\, z_{max}^{-\zeta_{IMF}}\, \Gamma_{\zeta_{IMF}}\left(-\frac{t}{t_{dep}}\right) \tag{5}$$

Here, $\Gamma_k(x)$ is the lower incomplete gamma function (subsuming a factor of $(-1)^{\zeta_{IMF}}$ that enforces reality), which tends to $x^k/k$ for small $x$ and $x^{k-1}e^x$ for large $x$. From this, it will be seen that the ratio is nearly 0 at early times, when only type II supernovae are operational, and tends toward infinity at late times, when star formation has effectively ended but type Ia supernovae are still exploding.

We then invert this expression to find the fraction of stars below a given metal-to-rock ratio:

$$f(R) = 1 - \exp\left(-\Gamma_{\zeta_{\text{IMF}}}^{-1}\left(0.053\, R\, \frac{\kappa^{0.81}\,\alpha^{0.56}}{\beta^{0.82}\,\gamma^{0.54}}\right)\right) \tag{6}$$

This fraction ranges from 0 for small $R$ to 1 for large $R$. The coefficient has been normalized so that the fraction of stars with $R$ greater than solar is 0.88 [46] (also in agreement with [45]), we find that the various candidate thresholds alter the probabilities of measuring our values of the constants as displayed in Table 2.

**Table 2.** The probabilities of observing the values of our physical constants for various metal-to-rock thresholds. All utilize the entropy and yellow habitability criteria. For a habitability criterion to be considered consistent with the multiverse hypothesis, all probabilities must be reasonably close to 1.

| Restriction | Reasoning | $\mathbb{P}(\alpha_{obs})$ | $\mathbb{P}(\beta_{obs})$ | $\mathbb{P}(\gamma_{obs})$ | $\mathbb{P}(\delta_{u\ obs})$ | $\mathbb{P}(\delta_{d\ obs})$ |
|---|---|---|---|---|---|---|
| None | baseline | 0.423 | 0.273 | 0.234 | 0.167 | 0.494 |
| $R < 1.9$ | outgassing | 0.262 | 0.042 | 0.461 | 0.183 | 0.327 |
| $R < 8$ | plate tectonics | 0.343 | 0.120 | 0.339 | 0.175 | 0.407 |
| $R > 0.08$ | plate tectonics | 0.428 | 0.276 | 0.225 | 0.168 | 0.498 |

As mentioned above, we take what we designated as the entropy and yellow habitability criteria in [11] as a baseline. We evaluate the compatibility of our additional habitability criteria by comparing their probabilities to those of the baseline. Generically, we say that a habitability criterion is disfavored by the multiverse hypothesis if the probabilities are significantly lower, and favored if the probabilities are significantly higher. If we take $R < 1.9$, as suggested by the outgassing criterion of [30], the probability of observing our value of the electron-to-proton mass ratio dips to 4.2%. If all probabilities are treated as independent, this criterion has a Bayes factor (ratio of product of probabilities) 7.3 times smaller than the baseline scenario. Though in the standard terminology of [47] this gives 'substantial' reason to expect the outgassing condition to not be important for habitability, the evidence gained if it is would not be enough to exclude the multiverse to any significance, since the chance of one of the probabilities being this small is only about one in five. The other two considerations have even less of a bearing on the probabilities. Even if the lower bound on $R$ is taken to be arbitrarily close to our value, the probabilities are affected by at most 50%, because this only excludes regions of parameter space that are not particularly fecund anyway. While one could say that the multiverse gives better odds for high metal planets being habitable, we conclude that the selection for a prime metal-to-rock ratio does not seem to be a determining factor for why we live in this universe. Likewise, the multiverse does not give us very strong expectations for which of these habitability criteria are correct.

## 3. Organics and Rocks
### 3.1. What Carbon-to-Oxygen Ratio Is Required for Life?

The life-giving properties and ubiquity of carbon have long been a fixture of modern anthropic arguments. A large part of [48] is devoted to extolling carbon's natural ability to form large biomolecules, dissolve in water, partake in multiple phases, and act as a mild acid. Today it is recognized as the key element comprising molecules important to living organisms, and defines organic chemistry. There is a long history of debate among astrobiologists over whether carbon is essential for life [49], or whether life may be based on some other substrate elsewhere in the universe [50,51] that continues to this day (see e.g., [52] for a recent review). Since the relative amounts of these elements can vary quite strongly depending on the constants, it is worthwhile to consider whether taking a stance on the necessity of carbon affects the probabilities of observing our universe's physical constants, and thus whether multiverse expectations make a prediction about its importance.

A major spur in the anthropic discussion of carbon's importance was Hoyle's 1954 observation that stellar carbon production is several orders of magnitude greater than naive expectations due to an improbable coincidence in the energy levels of nuclear states [53]. This coincidence causes a near equivalence between the energy of three helium nuclei and an excited level of the carbon-12 nucleus, which enhances the rate of this unusual fusion pathway, dubbed the triple-alpha reaction. Modern measurements put this energy difference at 380 keV [54], an impressively small number, given that it results from the sum of several terms all of the order 10 MeV. Due to this, the carbon abundance is seven orders of magnitude larger than the beryllium abundance, despite the trend for heavy elements to be less abundant than lighter ones. Equally important is the fact that the reaction to fuse oxygen from a carbon and a helium is slightly endothermic, so that the carbon initially created is not immediately destroyed [55]. Since this is controlled by nuclear binding energies as well, there is only a tiny region of parameter space where both carbon and oxygen exist simultaneously [56], and we happen to be situated in it.

This amount of fine tuning has been taken as evidence for the necessity of a highly specific C/O ratio, as otherwise it would be a strong coincidence that we find ourselves in such an apparently anomalous universe [57]. However, this has not yet been incorporated into our framework, where one tries to tally up the number of observers per universe for different habitability hypotheses regarding the importance of this ratio to determine which are compatible with our measurements of the physical constants. We undertake this analysis here, and also show how the traditional fine tuning arguments can be subsumed into our analysis.

For the dependence of the C/O ratio on the physical constants, we use the dependence of the various abundances calculated in [58] on $E_R = E_{12}^* - 3E_4$, the Hoyle resonance energy. There they modify this value in stellar evolution codes and keep track of nuclear interactions out to silicon, which allows us to study several relevant abundance ratios. These are displayed in Figure 1, which is reconstructed from Figure 9 of [58]. Note that this represents the yield averaged over solar metallicity supernovae, and does not take other processes, such as yields from medium mass stars, into account. As such, it represents the current best estimate for the dependence of these abundances on the constants, but it will not reflect the dependence exactly. The overall conclusions we draw from using this approximation are expected to hold, barring some unforeseen reason why the contributions from the neglected processes act to perversely undo these changes.

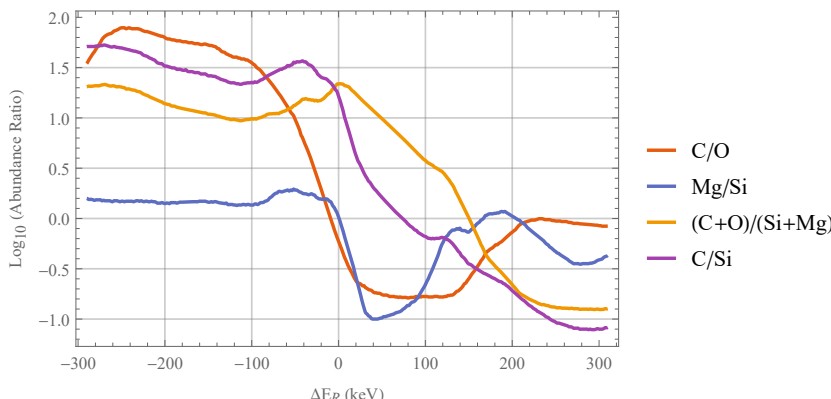

**Figure 1.** Abundance ratios of various elements as a function of the Hoyle resonance energy. The quantity $\Delta E_R = 0$ for our values of the constants.

To relate the change in the Hoyle resonance to physical constants, we use the relations found in [54] to arrive at:

$$E_R = \left(0.626\left(\delta_u + \delta_d\right) + 0.58\,\alpha - 0.004204\right)m_p \tag{7}$$

This is a lowest order Taylor expansion, but will be sufficient for our purposes, since the relevant thresholds are so close to our observed value. By combining this expression with the curves in Figure 1, we can determine thresholds on these parameters for any given ratio threshold.

The C/O ratio strongly affects planetary systems, so there is good reason to suspect it may have an effect on habitability. Its effects on planetary composition are very nearly stepwise around the value C/O = 1. This is a consequence of the fact that essentially all available C and O in the protoplanetary disk will combine to form carbon monoxide (CO); this leaves behind only the more abundant of the two to interact with other compounds [59]. For the solar value, C/O = 0.55 [60], which is very near the cosmic value of 0.6, the surplus oxygen forms silicates and oxides. If carbon were the more abundant element, carbide rocks like SiC and $Fe_3C$ would condense instead [59]. These are much sturdier rocks than silicates, are more resistant to weathering, and may not support plate tectonics as easily. These planets would have crusts made of graphite rather than quartz and feldspar [61], and may have oceans of hydrocarbons or tar rather than water [62].

Likewise, we may consider that universes with much lower C/O ratio would have correspondingly less material available for the construction of organic compounds, resulting in reduced biomass. A particular subject of study has been hydrogen cyanide (HCN), a key constituent of nucleic acids, and the elementary constituent of most, if not all, proposed prebiotic RNA synthesis pathways [63]. Primordial production of this compound was discussed in [64], where it was found that HCN is not produced in any appreciable abundance below C/O = 0.5, where $CO_2$ becomes the overwhelmingly dominant atmospheric constituent, and $CH_4$ is essentially absent.

It should be noted that stellar values of C/O need not necessarily correspond to the cosmic mean, and planetary values need not correspond to their host stellar values. The C/O ratio increases toward the galactic center [65], and also with time [62], though the enrichment timescale is several times the current age of the universe, and robustly longer than the star formation epoch. As a consequence, some stars in our universe have C/O > 1 [66]. The condensation of CO and $CO_2$ beyond their respective ice lines in the disk significantly affects the abundances of the bodies that form at that particular location; between the $H_2O$ and CO snow lines the C/O ratio can be significantly depleted from the stellar value, [67]. Chemical and disk evolution can further complicate the relation between planetary and disk C/O [68]. Atmospheric C/O is highly dependent on the planet's stochastic accretion history and subsequent evolution [69]. Lastly, local planetary C/O ratios can be substantially altered due to volcanic sources [70]. Here, we restrict our attention to planets inside the water snow line, and do not take variation about the cosmic mean into account. Though potentially important, the amount of scatter is dictated by galactic more than microphysical processes, and so a proper treatment of this will necessarily be delayed until we fold these galactic effects into our analysis, and vary cosmic parameters as well.

We can now test how the habitability hypothesis that carbon-rich systems are uninhabitable fares with multiverse reasoning. Using the ratio C/O = 1 to restrict our attention to silicate planets, which corresponds to $\Delta E_R = -8.9$ keV, we have the following probabilities of measuring our observed values of the constants:

$$\mathbb{P}(\alpha_{obs}) = 0.159, \ \mathbb{P}(\beta_{obs}) = 0.192, \ \mathbb{P}(\gamma_{obs}) = 0.251, \ \mathbb{P}(\delta_{u \ obs}) = 0.211, \ \mathbb{P}(\delta_{d \ obs}) = 0.371 \quad (8)$$

As can be seen, these are all extremely reasonable values, and do not differ significantly from the results we get when this criterion is not assumed. We may also include a lower bound on C/O: if we take the threshold to be 0.5 to ensure the production of HCN, corresponding to $\Delta E_R = 3.7$ keV, the probabilities change by at most 40%:

$$\mathbb{P}(\alpha_{obs}) = 0.292, \ \mathbb{P}(\beta_{obs}) = 0.213, \ \mathbb{P}(\gamma_{obs}) = 0.280, \ \mathbb{P}(\delta_{u \ obs}) = 0.228, \ \mathbb{P}(\delta_{d \ obs}) = 0.300 \quad (9)$$

Therefore, considering only these observables does not give us strong reason to expect this ratio to be important one way or the other.

However, we may augment these standard probabilities with additional measures. One observable corresponds to the probability of measuring our value of $E_R$. Note that this is equivalent to considering the probability of measuring C/Si to be at least as large as our value. The probability without restricting to any C/O range is $\mathbb{P}(E_{R\,obs}) = 0.456$, while including an upper bound of 1 leads to $\mathbb{P}(E_{R\,obs}) = 0.063$, including a lower bound of 0.5 leads to $\mathbb{P}(E_{R\,obs}) = 0.006$, and including both yields $\mathbb{P}(E_{R\,obs}) = 0.301$. The lower bound fares poorest, indicating that if HCN production is important, then silicate planets have to be as well.

Another quantity we can consider is the 'organic-to-rock ratio', (C+O)/(Si+Mg). As can be seen in Figure 1, we are situated very close to the peak of this quantity, so that our universe is exceptionally rich in organic material. If we consider the probability of measuring such a high value of this quantity without assuming that a particular composition is necessary for habitability, we find $\mathbb{P}((C+O)/(Si+Mg)_{obs}) = 1.2 \times 10^{-4}$, a highly unlikely value. However, if we restrict to universes with $0.5 < C/O < 1$, we instead find $\mathbb{P}((C+O)/(Si+Mg)_{obs}) = 0.287$. This is one way of formalizing the fine tuning argument that is often discussed regarding the triple-alpha process: if habitability does not depend much on the chemical content of the universe, then it is very coincidental that we happen to reside in a universe with such high organic content. However, unlike the standard anthropic argument, which only takes note of the fine tuning, our account of the relative probabilities of measuring this quantity ascribes a definite statistical significance to this coincidence. Correspondingly, this sets the significance by which the multiverse would be disfavored if this prediction is found to be wrong, and life is insensitive to organic-to-rock ratio. These results are summarized in Table 3. Lastly, note that hydrogen, the other key organic element, is the most abundant element in the universe by many orders of magnitude, and this fact does not change with the parameters we vary (though see [71] for a discussion on how the presence of hydrogen places constraints on the weak scale).

**Table 3.** The probabilities of our observations for various habitability criteria described in the main text. Each row indicates a purported habitability criterion (or combination), the associated restrictions on elemental abundance ratios, and the range of Hoyle resonance energies compatible with those restrictions. $\mathbb{P}(E_{R\,obs})$ is the probability of observing the Hoyle resonance energy to be at least as small as our measured value, and $\mathbb{P}((C+O)/(Si+Mg)_{obs})$ is the probability of measuring the 'organic-to-rock' ratio to be at least as large as our measured value. Probabilities greater than 0.1 are displayed in bold. All probabilities are computed utilizing the entropy and yellow habitability criteria.

| Reasoning | Thresholds | Range (keV) | $\mathbb{P}(E_{R\,obs})$ | $\mathbb{P}\left(\frac{C+O}{Si+Mg}\big|_{obs}\right)$ |
|---|---|---|---|---|
| baseline | none | - | **0.456** | 0.00012 |
| silicates | C/O < 1 | $-8.9 < \Delta E_R$ | 0.063 | 0.026 |
| HCN | 0.5 < C/O | $\Delta E_R < 3.7$ | 0.006 | 0.006 |
| silicates + HCN | 0.5 < C/O < 1 | $-8.9 < \Delta E_R < 3.7$ | **0.301** | **0.287** |
| olivine | 0.7 < Mg/Si | $\Delta E_R < 6.2$ | 0.010 | 0.006 |
| plate tectonics | 1 < Mg/Si | $\Delta E_R < 0.73$ | 0.001 | 0.001 |
| mantle viscosity | Mg/Si < 1.5 | $-83.9 < \Delta E_R$ | **0.404** | 0.017 |
| plate tectonics + mantle viscosity | 1 < Mg/Si < 1.5 | $-83.9 < \Delta E_R < 0.73$ | 0.007 | 0.007 |
| silicates + olivine | C/O < 1, & 0.7 < Mg/Si | $-8.9 < \Delta E_R < 6.2$ | **0.410** | **0.243** |
| silicates + plate tectonics | C/O < 1 < Mg/Si | $-8.9 < \Delta E_R < 0.73$ | 0.063 | 0.063 |

### 3.2. What Magnesium-to-Silicon Ratio Is Required for Life?

As with the C/O ratio, the initial Mg/Si of a planetary system has a large effect on the mineral content of its ensuing planets. As can be seen in Figure 1, this ratio is even more sensitive to the Hoyle resonance energy, and we are situated even more closely to an apparent boundary. Though the effects of the various purported thresholds on habitability

are still under active investigation, each can be incorporated into our analysis to generate predictions for which ones we expect to be important.

The cosmic Mg/Si ratio is 1.04, the spread is between 0.8 and 2 [72], and Earth's is 1.02 [73]. Major thresholds for this quantity are the values of 1 and 2; because the minerals that form from these either contain Mg and Si in 1:1 or 2:1 ratios, the relative abundance determines which rocks are formed in the protoplanetary disk, which species is left over to form other minerals, and the large scale properties of planetary interiors. For Mg/Si < 1, almost all magnesium is incorporated into pyroxene (given loosely by $MgSiO_3$), and the remaining Si forms silicates. Between 1 and 2, a mixture of pyroxene and olivine (loosely $Mg_2SiO_4$) form, as is the case for Earth. Above 2, and all Si is contained in olivine, with the remaining Mg in oxides such as periclase (MgO) [73]. Thus the Earth, and our universe, represent an intermediate phase between planets with excess magnesium and planets with excess silicon.

Planetary mantles which are dominated by these different minerals can have extremely different properties and behaviors. Mg acts very similarly to Fe in many mineral systems, and of the abundant elements, interacts most strongly with O in the primordial mantle [74]. The different magnesium compounds have different stereochemical structures, and so will have varying bulk properties. Pyroxene can be viewed as packs of long chains, making for a highly viscous mantle. Olivine, on the other hand, is comprised of packed silicate tetrahedra, leading to a less viscous mantle. Periclase is comprised of still smaller units, resulting in the most inviscid mantle of all three. Because mantle viscosity is exponentially sensitive to chemical binding energies, mantles dominated by periclase can be 100 times less viscous than our own [75]. This will have a profound impact on a planet's tectonics, but whether for the better or worse is unclear; Ref. [76] point out that such extreme volcanism would likely lead to an excess of methane, which could lead to a runaway greenhouse effect. Conversely, a low Mg/Si ratio can lead to a thermally stratified mantle that can arrest convection and just as significantly affect tectonic regime and outgassing rate [77]. A lower limit of Mg/Si < 0.7 was argued to lead to the absence of plate tectonics altogether in [72].

Thus the thresholds we have to consider are Mg/Si= 0.7, 1, and 2. The abundance ratio never exceeds 1.86, which occurs at $\Delta E_R = -59$ keV. The other two occur at 0.73 keV and 6.2 keV, respectively. We display these, along with the C/O thresholds discussed in the previous section, in Table 3.

Most rows in this table contain at least one anomalously small probability. Habitability criteria can only be said to be consistent with multiverse expectations if all probabilities are close to 1. Only by assuming that the C/O ratio lies within a narrow range do all probabilities exceed 10%. Taking the extreme stance that systems with Mg/Si < 1 are uninhabitable leads to exceedingly low probabilities, though this is alleviated to some extent if one also adopts the view that the C/O ratio is important. The less stringent bound of Mg/Si = 0.7, however, is perfectly compatible with these observations. This leads us to the following predictions: systems with marginally lower Mg/Si should be just as habitable as Earth, whereas systems with grossly different C/O should not be. If this is found not to be the case, this has the potential to cast severe doubt on the multiverse.

## 4. Additional Elements

On Earth, a wide range of elements are essential for biochemistry. These include the main constituents carbon, hydrogen, nitrogen, oxygen, phosphorus and sulfur, as well as trace metals such as iron. The ubiquity of these is controlled not only by which nuclides are produced in stars, but also which isotope is most stable for a given atomic number. This latter facet is just as sensitive to the physical constants as the production mechanisms, and because of the stepwise nature of nuclear stability, small changes to the values of the constants can lead to $\mathcal{O}(10-100)$ changes to the elemental abundances. In this section, we explore in detail how these stability thresholds affect several of the most important elements for life, namely nitrogen, phosphorus, sulfur, and the iron peak elements.

To investigate the conditions for nuclear stability, we use the semi-empirical mass formula, which is a phenomenological model that makes extrapolating nuclear properties to different values of the physical constants possible [78]. With this, the binding energy of a nucleus with $A$ nucleons and charge $Z$ is

$$E_b(A, Z) = a_v A - a_s A^{2/3} - a_c \frac{Z(Z-1)}{A^{1/3}} - a_{sym} \frac{(A-2Z)^2}{A} + \delta \tag{10}$$

The first terms are the volume and surface energies, which result from nearest neighbor strong interactions. The third term is due to Coulomb repulsion, the fourth is due to Fermi repulsion, and gives preference to nuclei with equal numbers of protons and neutrons, and the fifth encapsulates spin coupling, giving preference to nuclei with even numbers of protons and neutrons.

The dependence on $\alpha$ and $m_p$ of these coefficients are taken to be:

$$
\begin{aligned}
a_c &= 0.714 \frac{\alpha}{\alpha_0} \frac{m_p}{m_{p0}} \text{ MeV,} \\
a_{sym} &= 23 \frac{m_p}{m_{p0}} \text{ MeV,} \\
\delta &= 34 \frac{(-1)^Z}{A^{1/2}} \frac{m_p}{m_{p0}} \chi_{evens}(A) \text{ MeV}
\end{aligned}
\tag{11}
$$

The function $\chi$ is the indicator function, which equals 1 when $A$ is even and vanishes otherwise. We also take the difference between the proton and neutron masses to be given by [79]:

$$m_n - m_p = m_d - m_u - 1.21 \frac{\alpha}{\alpha_0} \frac{m_p}{m_{p0}} \text{ MeV} \tag{12}$$

This is used to determine which element $Z$ is the most energetically favorable for a given nuclear weight $A$ by considering the process of beta decay. This process can only occur if the quantity

$$\Delta E = E_b(A, Z) - E_b(A, Z-1) + m_n - m_p - m_e > 0. \tag{13}$$

The total abundance of a given element will the the sum of all the isotopes for which that element is stable, multiplied by the abundance of each of those isotopes. In Figure 2, we display the abundances of the second and third row elements according to this criterion.

In this figure, we have kept track of isotopes up to atomic number 86, and have used the abundance values given in [80], who record abundances for each nuclear weight $A$. It is these values which are most appropriate for our purposes, as they are insensitive to the physical constants and can be treated as fixed. The elemental abundances are then calculated by grouping these abundances according to which element $Z$ is energetically preferred for every $A$. In general, these thresholds will occur along hyperplanes in parameter space of the form $\beta + \delta_d - \delta_u = k\alpha + c$, where $k$ ranges from 0 to 2, with an average over the first 40 isotopes of 1.005. To display the extent of this effect, then, we plot it in the maximally sensitive direction, where the change in $-\beta - \delta_d + \delta_u$ is proportional to the change in $\alpha$. This is given by the dotted line in Figure 3. Bear in mind, however, that restricting to this subspace is only a first order approximation, and the details of Figure 2 can be stretched out in various directions of parameter space.

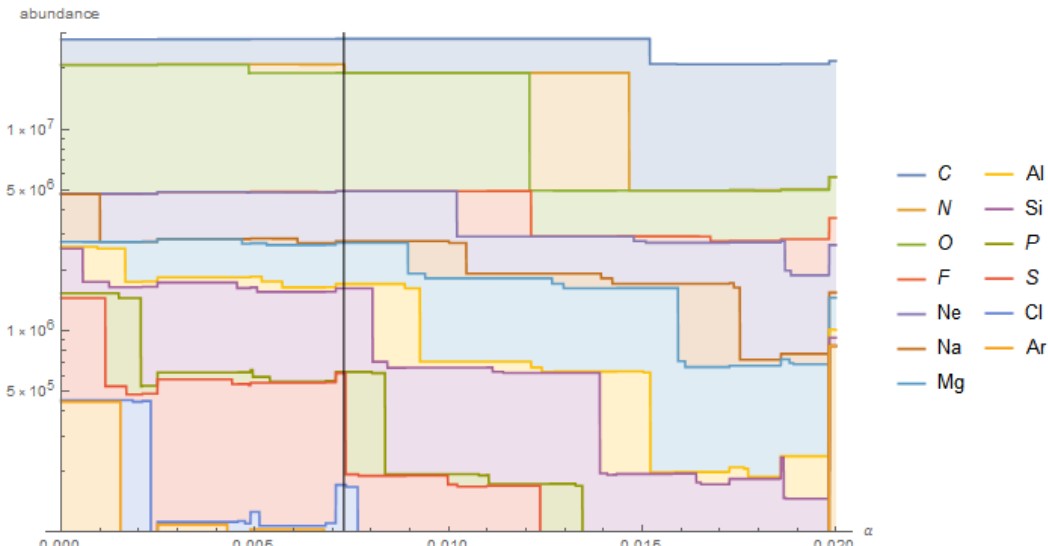

**Figure 2.** The abundances of second and third row elements, restricted to the maximally sensitive direction in parameter space described in the text, and parameterized by $\alpha$. The black line denotes our observed values. The elements are ordered from the upper right downward.

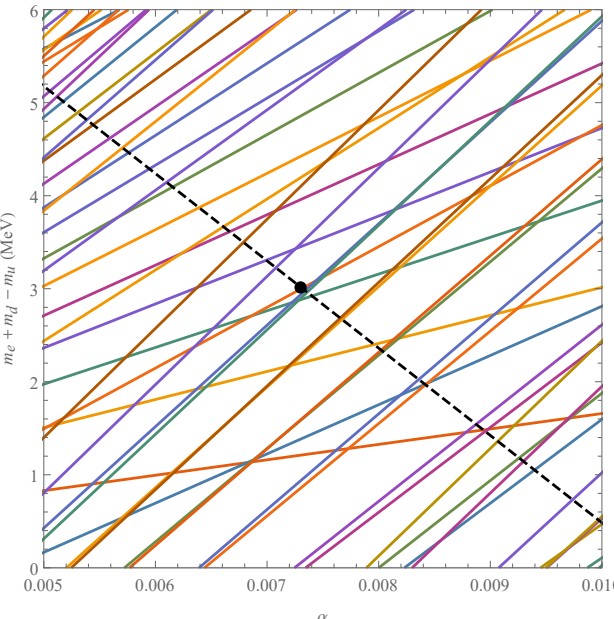

**Figure 3.** Nuclear stability thresholds for elements from the first three rows of the periodic table. The black dot represents our values. The black dashed line is the maximally sensitive direction, averaged over these isotopes. Each domain in the figure has fixed abundance ratios, and the thresholds correspond to the abrupt changes displayed in Figure 2.

Several features of these plots should be remarked upon. First is the fact that many transitions are present, beyond which nearly the entirety of a particular element is swapped out for its neighbor. These transitions are abrupt, so that there are stretches of parameter space where a particular element may be nearly absent, but beyond which conditions are changed to the point where that element is produced as a completely different isotope. Notice that the abundances are organized into several horizontal stripes in Figure 2 representing the different nuclear weights $A$, and as $\alpha$ increases, the preferred charge $Z$ decreases stepwise. The overall trend for larger $\alpha$ is to produce universes with less heavy and more light elements. An exception to this can be seen at the very right edge of the plot, where the argon abundance spikes up. This occurs because the stability of the iron peak, $A = 56$, has

traversed to the point where it has entered the third row of the periodic table, which we will discuss more thoroughly below. Even charges are generically stable for larger regions of parameter space than odd charges due to the pairing term.

Because there are so many relevant isotopes, and there are up to a dozen such transitions for each isotope as parameters are varied, hundreds of such transitions occur within the range of several times the observed value of $\alpha$. As such, we expect some transition to occur whenever the constants are varied by even a fraction of a percent. The result is a highly heterogeneous checkerboard of elemental abundances throughout parameter space, producing universes which can drastically differ with regards to certain, possibly key properties.

Before we go on to consider several specific elements, let us state a few caveats in this analysis. As stated before, we normalize the baseline abundances to those in our universe. As explored in previous sections, these abundances may change if the yields of massive stars, or the ratio of yields between differing supernovae, are altered. Our analysis in this section can be seen as complementary to these discussions; this is in part to disentangle the origins of these different effects, and in part because a full account, incorporating all three separate effects, is a much more ambitious undertaking, that will require calculating the parameter dependence of the supernova yield of every element. Additionally, we do not consider instability due to electron capture, and do not treat cases where unstable elements have half life longer than the age of the universe as effectively stable. These borderline cases will shift the thresholds somewhat, but not appreciably.

With these preliminaries, we can now specialize our discussion to several nearby thresholds, and the consequences each has for the abundances of some of our most cherished elements.

### 4.1. Is Nitrogen Essential for Life?

Nitrogen is essential for life as we know it, being a constituent of proteins, DNA, and RNA. It is the linchpin of the peptide bond, which forms the structural backbone of proteins. The reason this has been selected by biochemistry over other potential scaffolds seems to be nitrogen's ability to form three bonds, which allows it to bind two molecules together while maintaining a positive charge [81]. This increases its stability in aqueous solution and its affinity for nucleic acids. Nitrogen's triple bond is not wholly beneficial for life, though, as it results in the majority of nitrogen atoms pairing off, rendering it a mostly inert gas. As such, nitrogen is often a limiting nutrient in ecosystems [82]. Still, one may be able to convince themselves that microbial life, and the complex life it supports, would do just as fine in nitrogen poor universes by prioritizing fixation [83] and/or tightening cycling of bioavailable N chemical species [84]. However, without a noncondensible gas in the atmosphere, liquid surface water would be much less stable on terrestrial planets [85].

Despite its ubiquity, nitrogen is actually one of the most precariously stable element in our universe. The vast majority of nitrogen-99.6%-is nitrogen-14, produced during the CNO cycle of massive stars, with some contribution from smaller stars as well [86]. Its binding energy is extremely small; from the semi-empirical mass formula, Equation (13):

$$\Delta E\left({}^{14}\text{N}\right) = -0.0017\, m_p + 0.696\, \alpha\, m_p - m_e - m_d + m_u \tag{14}$$

This equates to just 156 keV for our values[5].

The flip side of such marginal stability is the fact that in our universe, carbon-14 decays with extremely tiny reaction energy. The primary consequence of this is a drastic increase of its half life by a factor of $10^6$, making it by far the lightest element that decays on kyr timescales. This can be attributed to an accidental cancellation between density-dependent nuclear forces [87]. The tuning is quite significant- if $\alpha$ were increased by just 3.4% nitrogen-14 would decay into carbon-14, leading to a cosmic nitrogen abundance reduced to a factor of 0.0037 the observed value.

In light of the importance of nitrogen, we may consider the hypothesis that universes with such significant reduction in nitrogen are uninhabitable. Adopting this stance leads to the following probabilities:

$$\mathbb{P}(\alpha_{obs}) = 0.049, \; \mathbb{P}(\beta_{obs}) = 0.071, \; \mathbb{P}(\gamma_{obs}) = 0.163, \; \mathbb{P}(\delta_{u\;obs}) = 0.080, \; \mathbb{P}(\delta_{d\;obs}) = 0.182 \quad (15)$$

A comparison to the baseline in Equation (1) (and again treating each probability as independent) leads to a Bayes factor of 268 lower for this criterion, meaning that, if nitrogen is taken to be essential for life, the probability of our observations would be lowered by this amount compared to the baseline case. Evidently, this expectation is comparatively quite highly disfavored from multiverse expectations. This may be somewhat surprising, given the many important roles nitrogen plays. We hesitate to conclude that this is incompatible with the multiverse, however, since all probability values are not unreasonably small.

To compare these two hypotheses, the recommendation is to search for life in nitrogen poor regions of our universe. If atmospheric nitrogen is the result of tectonic forces, we may look at super-Earths which are too large to support plate tectonics [88], or, if it is the result of influx from the outer system, we may look at planets that have been subjected to less late stage delivery [89]. If we indeed find that these environments are unfit for life (controlling for other concomitant factors), this would run counter to our predictions above, and so count as a strike against the multiverse hypothesis.

We note that if the C/O habitability criterion is included in conjunction with the N criterion, the probabilities are significantly ameliorated. This will be discussed further in Section 5.

### 4.2. Are Phosphorus and Sulfur Essential for Life?

The most utilized third row elements in biochemistry are phosphorus and sulfur. The main utility of these third row elements is that they are capable of forming additional chemical bonds. Phosphorus is particularly important for life, phosphate ($PO_4$) being a key component of DNA, RNA, ATP, and phospholipids. These essential roles derive from several properties of phosphorus: first, it has the ability to link multiple atoms while maintaining a (single) negative charge, which inhibits reactions and promotes its stability. Second, phosphate is kinetically unstable and so can act as an energy source, but this process is thermodynamically unlikely without an enzyme, allowing its controlled release [90]. Phosphorus is often a limiting nutrient [91], and an increase in the global phosphorus cycle has been argued to be the cause of the Cambrian explosion [92]. In [93], phosphorus abundance was argued to have a strong effect on planetary habitability.

Some hypothetical biochemistries have been put forward that would replace phosphorus with arsenic, though whether this would ultimately be suitable remains to be seen. As has been pointed out for example in [94], arsenic is much less stable in water, and maintaining the molecular configuration of an arsenic-containing biomolecule represents a significant challenge. Additionally, being a fourth row element, the abundance of arsenic is 1000 times lower than phosphorus.

Essentially all phosphorus and sulfur are produced during oxygen burning in massive stars [95], the resultant isotopes being $^{31}$P and $^{32}$S. Since sulfur is an alpha element, it is a factor of 53 times more abundant than phosphorus in our universe.

However, this situation will be reversed if sulfur-32 is unstable. It will decay to phosphorus-32 if the energy difference from Equation (13) is negative:

$$\Delta E\left(^{32}S\right) = -0.005\, m_p + 1.16\, \alpha\, m_p - m_e - m_d + m_u$$

This is actually the closest nuclear stability threshold (of the cases we consider here). This threshold is crossed if $\alpha$ is increased by 0.7%, at which point the phosphorus abundance is increased by a factor of 51.5, and the sulfur abundance is decreased by a factor of 0.049.

This region of enhanced phosphorus and decreased sulfur does not go on indefinitely, however. As can be seen from Figure 2, after a certain point chlorine-35 becomes unstable

and will decay into sulfur, and beyond that phosphorus-32 will decay to silicon. These thresholds are given by

$$\Delta E\left(^{35}\text{Cl}\right) = 0.00565\, m_p + 1.197\, \alpha\, m_p - m_e - m_d + m_u$$

and

$$\Delta E\left(^{32}\text{P}\right) = -0.0070\, m_p + 1.096\, \alpha\, m_p - m_e - m_d + m_u \tag{16}$$

The first of these thresholds is crossed at a 5.0% increase of $\alpha$ and results in a universe with 0.25 the amount of sulfur as ours, and 0.01 the amount of chlorine. The second is crossed with an increase in $\alpha$ of 14.6%, and results in a universe with 0.40 times the amount of phosphorus as ours.

There are a number of different stances one may take on the habitability of these various regions. If we take the habitability of a location to be proportional to the amount of phosphorus it contains, then the neighboring region of parameter space would be 50 times more habitable than our universe. With this ansatz, the probabilities of observing our constants become

$$\mathbb{P}(\alpha_{obs}) = 0.076,\ \mathbb{P}(\beta_{obs}) = 0.482,\ \mathbb{P}(\gamma_{obs}) = 0.338,\ \mathbb{P}(\delta_{u\ obs}) = 0.299,\ \mathbb{P}(\delta_{d\ obs}) = 0.088 \tag{17}$$

These values are disfavored compared to the baseline by a Bayes factor of 6.8, suggesting that habitability should not be proportional to phosphorus content. This expectation is not unreasonable, given that on Earth phosphorus, while being a limiting nutrient, is recycled on average some 500 times before exiting an ecosystem [96]. In principle this could be even higher on planets where phosphorus is even more limited. This could be checked by searching for signs of life around stars that have anomalously low phosphorus content, as described for instance in [97].

Alternatively, one could worry that universes with decreased sulfur abundance would be detrimental to life. In this case, the probabilities become

$$\mathbb{P}(\alpha_{obs}) = 0.436,\ \mathbb{P}(\beta_{obs}) = 0.285,\ \mathbb{P}(\gamma_{obs}) = 0.250,\ \mathbb{P}(\delta_{u\ obs}) = 0.163,\ \mathbb{P}(\delta_{d\ obs}) = 0.494 \tag{18}$$

These are all within 4% of the baseline values, making the probability of our observations almost independent of whether a dearth of sulfur is detrimental to life or not. As such, the multiverse does not provide any strong predictions for this habitability criterion.

*4.3. Is Iron Essential for Life?*

Differing values of the constants will lead to different end products of the iron peak. While this does not lead to any strong predictions, we include it here for completeness, and because it may be of some passing interest to the reader.

Stellar fusion will always stop at nickel-56; this isotope is 'doubly magic', leading to an enhanced binding energy. In our universe, this nickel subsequently undergoes two beta decays to ultimately yield iron. The energetically preferred isotope varies with the constants, however. This dependence was initially explored in [98], though attention was restricted to $\alpha$, and no attempt at inferring predictions for habitability was made. Generically, the endpoint element $Z$ can be computed from Equation (13) to be

$$Z_{56} = \text{floor}\Big[\big(\{\tfrac{26.54}{25.94}\} - 0.23(\hat{\alpha} - 1) + 0.14(\hat{\beta} - 1)$$
$$+ 1.27(\hat{\delta}_d - 1) - 0.59(\hat{\delta}_u - 1)\big)/(1 + 0.10(\hat{\alpha} - 1))\Big] \tag{19}$$

where the top coefficient is relevant for even $Z$, and the bottom is relevant for odd $Z$. We use the notation $\hat{x} = x/x_{obs}$. Figure 4 displays the abundances of the fourth row elements, which are primarily dictated by the location of this peak. One can see that for different values of the constants, essentially all the iron on Earth would be replaced by another

element. For nearby values these would be another transition metal, and so may not produce such a drastic change. We will leave it to the reader to imagine how different the world would be if iron were replaced with potassium, or beyond that the noble gas argon. The general trend is that the higher $\alpha$ is, the smaller the elemental constituents of the universe are. For $\alpha$ beyond 3 times its observed value, essentially all fourth row elements will be absent.

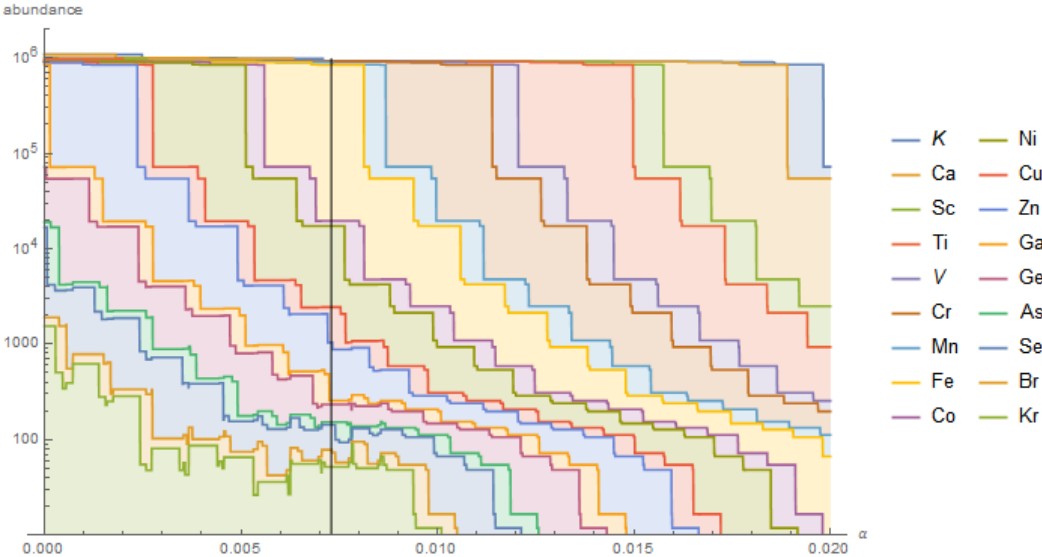

**Figure 4.** The abundances of the fourth row elements, restricted to the maximally sensitive direction in parameter space described in the text, and parameterized by $\alpha$. The black line denotes our observed values. The elements are ordered from the upper right downward.

The closest thresholds are as follows: if $\alpha$ is increased by 11.4% (again, restricting to the line in parameter space which is maximally sensitive to the change in constants), the iron abundance is decreased by a factor of 0.067, replaced instead by manganese. An increase of 19.1% would result in chromium, which would be enhanced by a factor of 68. A decrease in $\alpha$ by 23.3% would lead to a cobalt universe, and a decrease by 29.8% would make nickel stable.

The most conservative hypothesis would be to take universes for which iron is not the end product of this decay chain as uninhabitable. This stance is bolstered by the recent finding that iron may have been an essential component for many of the most essential reactions in biochemistry [25]. Taking iron to be essential, the values of the probabilities become

$$\mathbb{P}(\alpha_{obs}) = 0.303, \ \mathbb{P}(\beta_{obs}) = 0.312, \ \mathbb{P}(\gamma_{obs}) = 0.289, \ \mathbb{P}(\delta_{u\ obs}) = 0.318, \ \mathbb{P}(\delta_{d\ obs}) = 0.247 \ (20)$$

These are not substantially different from the probabilities that are completely agnostic to the exact metal present. Similar probabilities result for any metal-specific habitability criterion one may adopt (any metal above chromium, for instance). Therefore, no predictions regarding the habitability of different metals can be made.

As a final note, let us comment on the absence of other key elements in our discussion, namely sodium and chlorine. The sodium abundance is relatively stable across parameter space compared to other elements, and the closest thresholds actually increase its abundance rather than decrease it. Chlorine-36 does become unstable if $\alpha$ changes by +9.1% (−12.6%), resulting in a decrease in abundance by a factor of 72 (23), but these thresholds are far enough away that they do not induce a significant change in any of the probabilities.

## 5. Discussion

In a multiverse setting, many of the properties of our universe are not unique, but instead could conceivably have been different. We expect some range of conditions to be habitable, and that our experiences are not too atypical. This leads us to the conclusion that we should probably inhabit a universe that makes a relatively large number of observers. Since the number of observers that can exist within a universe is highly dependent on the assumptions we make about the requirements for complex life, only some of these habitability assumptions will be compatible with these generic multiverse expectations.

It is tempting to use this reasoning to explain every unique facet of our universe. Since our universe has unusually high abundances of many of the elements which are principle components of living systems, for instance, we might like to conclude that universes with different elemental palettes are sterile. Such reasoning is preemptive, however, and dangerously circular; are we made out of CHNOPS because these are the only elements capable of comprising biochemical compounds, or simply because they are the most abundant? Surely not every feature of our universe would be tuned to be maximally conducive to life, and typical observers ought to experience several facets of their surroundings to be suboptimal. Especially since we have found that very abrupt changes to the cosmic abundances are common, a more thorough investigation of the impact of each habitability assumption is necessary before any conclusions can be drawn.

Throughout, we investigated three sources of abundance variations: the relative outputs of the two primary supernova sources, the final yield dependence on nuclear resonances, and the changes to the stable isotopes. All three effects were found to produce changes that can profoundly alter the basic structure of the macroscopic world, sometimes when our physical constants are varied by only a fraction of a percent. All this begs the question as to whether we really expect our minute region of parameter space to contain the only recipe capable of producing life, or whether universes are just starkly diverse, and each of them contain observers marveling at how perfectly theirs is suited to their own brand of life.

The answer we have found lies somewhere in between these extremes. Some thresholds truly should be significant, or else our presence in this universe would be highly atypical. However, the majority of thresholds we consider, when treated as significant, were either shown to have very little effect on the probability of our observations, or else decrement them. This conclusion was only reached by considering a wide range of potential features: the metal-to-rock ratio (Table 2), the carbon-to-oxygen ratio, the magnesium-to-silicon ratio, the organic-to-rock ratio (all Table 3), the nitrogen abundance (Equation (15)), the phosphorus abundance (Equation (17)), the sulfur abundance (Equation (18)), and the iron abundance (Equation (20)). Of these eight potentially important habitability criteria, only the carbon-to-oxygen ratio was determined to be important to very high significance. Even though the others are affected by nearby thresholds within parameter space, ascribing special significance to these was found to be erroneous.

As was the explicit intent of this exercise, these findings have implications for our expectations for which regions in our universe we expect to be habitable: planets with more metals should be habitable. Carbon poor (or rich) planets should not. Highly magnesium rich planets may or may not be habitable, but mildly magnesium rich planets should be habitable. Nitrogen and phosphorus poor planets should be habitable. If any of these predictions turns out to be wrong, then the majority of observers would have arisen in universes different from ours, and the multiverse would be ruled out to potentially very high statistical significance, depending on the case.

We should stress at this point that all these conclusions take the base model that habitability is proportional to the entropy processed by a planet, and that oxygenic photosynthesis is necessary for (or at the very least greatly facilitates) complex life. These are not the only base assumptions which are compatible with our observations, and we chose them for sake of narrativization only. This highlights a major shortcoming in our analysis thus far; since we have chosen a habitability criteria that 'works' as our baseline,

the addition of any other criteria is bound to be either insignificant or detrimental. There may equally well be habitability hypotheses from our initial list, dismissed at first because they are untenable on their own, which become viable when combined with one of these additional considerations. Additionally, we did not consider the 256 possibilities that result from including multiple abundance assumptions simultaneously. A fuller exploration is needed, and we remedy these oversights here.

When taken in conjunction with the criteria we considered in previous papers, it is becoming impracticable to compute all possible combinations. An exhaustive computation would thus far include 509,607,936 unique combinations, from 22 different habitability considerations, corresponding to over 100 CPU-years. This number grows exponentially with the number of factors; since we will include even more habitability criteria in the future, clearly some way of mapping this space of possibilities without going through every combination is needed. To do this, we neglect the habitability criteria that had no bearing on previous calculations (such as mass extinction rate and hot Jupiter rate), and furthermore we restrict our consideration to the presence of three or less criteria at a time. This gives fairly good coverage of effects that may occur due to the interaction of different boundaries, but admittedly does not fully sample the space of possibilities. However, this restricted computation scales cubically with the number of habitability criteria, and the number of combinations remains under 10,000 until 40 criteria are considered.

As a base, we consider the following habitability hypotheses for conditions required for complex life: (i): photosynthesis (optimistic and pessimistic bounds, denoted photo/yellow), (ii): tidally unlocked planets (TL), (iii): stellar lifetimes of several billion years (bio), (iv): plate tectonics (plates), (v): terrestrial mass planets (terr), (vi): temperate planets (temp), and that the probability of the emergence of life is proportional to: (vii): stellar lifetime (time), (viii): planet area (area), and (ix): entropy (entropy). To these, we add the additional criteria from this paper: that complex life requires: (x): a certain metal-to-rock ratio (metal), (xi): a certain C/O or Mg/Si ratio (C/O, Mg/Si), (xii): nitrogen, phosphorus, or sulfur (N,P,S), and (xiii): iron (Fe). These 13 conditions result in 756 combinations to check.

For each of these, we compute the probabilities of the five physical constants we have been considering, $\alpha$, $\beta$, $\gamma$, $\delta_u$, and $\delta_d$, as well as the probability of orbiting a star as massive as our sun, and of the observed organic-to-rock ratio. To give the reader a sense of which combinations are viable, we report all those for which all probabilities are greater than 0.01:

C/O + entropy + ( photo or yellow or TL)

C/O + area $\pm$ (photo or yellow or TL or bio or plates or terr or metal or N or P or S or Fe)

Mg/Si + area + N

The full list, with numerical values, is included in the code repository mentioned in the methods section below Section 6.

Several important lessons should be taken from this: in addition to the entropy + yellow branch we have been focusing on (which gives the overall highest probabilities), there is an additional branch centered around the area condition. In all but one viable criteria, a specific C/O ratio is required. In contrast to the cases when considered in isolation, the nitrogen, phosphorus, metal-to-rock ratio, and plate tectonics criteria are compatible habitability criteria when the area condition is important. This last point partially holds on the C/O + entropy branch as well, though these were not covered by this analysis which terminated at combinations of three criteria; the addition of the nitrogen and phosphorus criteria become compatible with our observations, but the plate tectonics and metal-to-rock ratio criteria remain disfavored.

A histogram of the smallest probability for the 756 combinations considered is displayed in Figure 5. Of the combinations considered, only in 2% do all values exceed 0.01, only 6% exceed 0.001, and 32% exceed 0.0001.

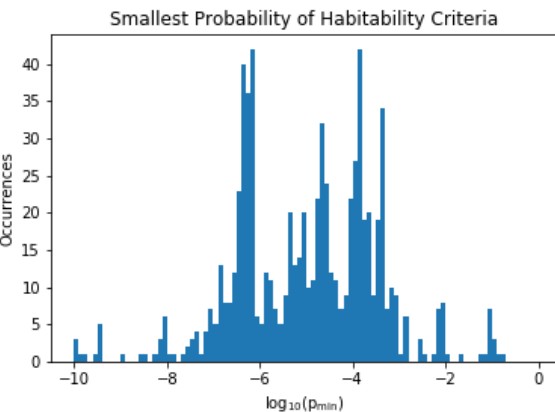

**Figure 5.** The smallest probability of the combinations of the 756 habitability criteria considered. All combinations of 3 or less criteria discussed in this paper and the previous four in this series are included. The probabilities considered are of observing the five physical constants being varied, as well as the probability of orbiting a star of solar mass and witnessing our organic-to-rock ratio.

This should be compared to the expected smallest value of seven random numbers, which has a cumulative distribution function of $1 - (1 - p)^7$. Thus, the fact that there are seven chances for one number to be unnaturally small decreases the significance of any value by that same factor, but this has little bearing on the majority of cases.

The majority of potential habitability criteria are incompatible with the multiverse, and so, if found to be true, would rule out the multiverse to a high statistical significance. The multiverse predicts relatively specific conditions that should be important to life, and equally predicts the unimportance of others. This goes a long way toward establishing the multiverse as a predictive, testable scientific framework. To the list of about a dozen predictions that have been made in previous parts of this series, we now add a few more, as well as recommendations for how to test each with upcoming (or more distant) future missions. Further work will yield even more predictions, maximizing both the chances and certainty of testing this framework.

## 6. Methods

In the previous papers of this series, we focused our attention on only three physical constants: $\alpha$, $\beta$, and $\gamma$, and held the others fixed. The chemical abundances, however, depend sensitively on the light quark masses, which necessitates the inclusion of these quantities in this paper. We briefly describe the changes this required to our numerical computations, and the updates to our previous conclusions this brings about. The full python code is available at https://github.com/mccsandora/Multiverse-Habitability-Handler, accessed on 2 December 2022.

Our code is built to numerically compute the probabilities quoted throughout the paper, of observing our universe's values of the 5 physical constants we vary, the probability of being around a sunlike star, the probability of observing such a small Hoyle resonance energy, and the probability of observing such a high organic-to-rock ratio. These are computed using the Monte Carlo integration technique, which generates random points throughout the range and equates the desired integral with the expectation value of the integrand. To generate the pseudo-random points, we use the Sobol sequence method, which avoids the clustering inherent in truly random processes, increasing accuracy [99]. The sample size is set to achieve our target accuracy of 1%, which is determined by randomly splitting the generated sequence and comparing the values of the two resultant probabilities.

While Monte Carlo methods are well suited for high dimensional integrals, the inclusion of the anthropic boundaries reduces the number of points utilized from a hypercube covering the relevant parameter space to 2%. Including the plate tectonics and/or C/O boundaries reduces this by a further 2 orders of magnitude each, making a brute force Monte Carlo approach untenable. For each of the four cases, we rescaled the initially

generated points to the relevant parallelotopes, enhancing efficiency and accuracy. The combination of the plate tectonics and C/O boundaries is particularly economical with this rescaling approach, using 30% of the generated points, because the conjunction of the two boundary regions restricts the constants to lie within a narrow range of parameter space.

The anthropic boundaries we include are those found in [100]. Stated briefly, they are as follows: (i) the proton and neutron should be the most stable nuclei, rather than the $\Delta^{++}$ or $\Delta^-$, (ii) heavy elements are stable, (iii) the proton is stable in nuclei, (iv) hydrogen is stable, both to positron emission and electron capture, (v) proton-proton fusion is exothermic, (vi) the deuteron is stable, both to strong and weak decays, and (vii) the diproton is unstable. Each of these boundaries have been contested [101–107], some by the present authors, but the exclusion of most individual bounds does not have a large effect on the probabilities we compute (this holds especially true for lower bounds). Because the intent of this paper is to generate potential tests of the multiverse based off probing regions of our universe that closely resemble normal environments within other universes, and because there are no regions of our universe that resemble a universe where any of these bounds do not hold, we do not gain anything by relaxing any of these bounds.

**Author Contributions:** Conceptualization, all authors; Methodology, M.S.; Formal Analysis, M.S.; Validation, V.A., L.B., G.F.L. and I.P.-R.; Writing—Original Draft Preparation, M.S.; Writing—Review and Editing, V.A., L.B., G.F.L. and I.P.-R. All authors have read and agreed to the published version of the manuscript.

**Funding:** This research received no external funding.

**Data Availability Statement:** All code to generate data and analysis is located at https://github.com/mccsandora/Multiverse-Habitability-Handler, accessed on 2 December 2022.

**Conflicts of Interest:** The authors declare no conflict on interest.

## Appendix A. Binary Evolution

Our calculation of the type Ia supernova rate required estimating $f_{\text{binary}}(a_{\text{Roche}})$, the fraction of binary stars whose separation is close enough for significant mass transfer. This can be estimated by considering the distribution of initial separations, the critical separation, here taken to be the Roche radius $a_{\text{Roche}}$, and a model for orbital evolution of the system, which accounts for effects that can significantly tighten the pair's orbit over the course of its evolution. Ultimately, the dependence of this quantity on physical constants will be shown to be very weak, but we include the calculation here for completeness.

The initial separations are found to be given by a log-uniform distribution, $f_{\text{binary-0}}(a) = 1/a/\log(a_{\text{max}}/a_{\text{min}})$, from [108]. Here, the maximum separation is given by the typical intracluster separation, $a_{\text{max}} \sim n_{\text{cluster}}^{-1/3}$, and the minimum separation is given by the separation at which the system becomes unstable to merging. This latter scale is given by the Roche limit, which scales as the stellar radius $a_{\text{min}} \sim R_\star$ [41]. A milder version of this criterion sets the separation needed for significant mass transfer in terms of the stellar radius as well, but multiplied by a factor of a few.

There are four sources of orbital decay that could potentially play an important role in determining the fraction of binary systems that become supernovae within a given amount of time. We will detail each of these in turn. The first comes from gas drag within the initial cluster [42]. This leads to an evolution given by

$$\dot{a} = -\frac{\xi_d\,G^3\,M_\star^2\,\rho}{c_s^5}, \quad a(t_{\text{cluster}}) = a_i - \frac{\xi_d\,G^3\,M_\star^2\,\rho}{c_s^5}\,t_{\text{cluster}} \tag{A1}$$

Here $q$ is the mass ratio of the system, $\rho$ is the cluster density, and $c_s$ is the gas speed of sound. Throughout this section, the $\xi$s represent $\mathcal{O}(1)$ quantities, many of which must be fit to simulations. As can be seen, this leads to a constant shift inward, independent of initial separation. Using the cluster lifetime $t_{\text{cluster}} \sim 1/\sqrt{G\rho}$ and the virial speed

$c_s \sim \sqrt{G\rho^{1/3}M_{\text{cluster}}^{2/3}}$, we find this shift to be parametrically of the order $\Delta a \sim a_{\text{max}}/N_{\text{cluster}}^{5/3}$, where $N_{\text{cluster}}$ is the number of stars in the cluster. This is always much smaller than the maximal separation, but significantly sculpts short period systems, causing the distribution to resemble a log-normal [42].

Another source of intracluster evolution comes from close encounters with other stellar systems. For systems whose binding energy exceeds the typical cluster energy, an encounter usually results in the system becoming tighter [109]. The hardening rate for this process is given by

$$\dot{a} = -\frac{\zeta_e \, G \, \rho}{c_s} \, a^2, \quad a(t_{\text{cluster}}) = \left( a_i^{-1} + \frac{\zeta_e \, G \, \rho}{c_s} \, t_{\text{cluster}} \right)^{-1} \tag{A2}$$

This contribution to orbital evolution becomes less effective for small separations, so, while it can lead to a skewing of the distribution, it is usually not directly responsible for coalescence.

A third component to orbital evolution comes from gravitational waves. This evolution is given by [110]

$$\dot{a} = -\frac{\zeta_g \, G^3 \, M_\star^3}{4 \, a^3}, \quad a(t) = \left( a_i^4 - \zeta_g \, G^3 \, M_\star^3 \, t \right)^{1/4} \tag{A3}$$

This effect is very weak throughout all of parameter space, however, and will only affect those systems with very small separations.

Finally, the most relevant contribution to orbital evolution is given by magnetic braking. For systems whose orbits are close enough for the stellar rotation to become orbitally locked, the magnetic drag from the star's differential rotation, which ordinarily causes rotational spindown, can backreact to cause orbital spindown as well [111]. Taking the rotational magnetic braking to be given by the usual Skumanich form, angular momentum loss is related to angular velocity through $\dot{J} = -\tau \, \Omega^3$. The resultant orbital evolution is given by

$$\dot{a} = -\frac{\zeta_s \, \tau \, G}{5 \, a^4}, \quad a(t) = \left( a_i^5 - \zeta_s \, \tau \, G \, t \right)^{1/5} \tag{A4}$$

What remains is to determine the origin of the parameter $\tau$ in the Skumanich formula: from [112,113], we can express the rotational spindown as $\dot{J} \sim U R_\star^2 \max(R_\star^2, R_{\text{Alfven}}^2)\Omega^3/G$, where $R_{\text{Alfven}}$ is the Alfven radius, and $U \sim \sqrt{T_\star/m_p}$ is the convective speed. Specializing to the regime where $R_\star > R_{\text{Alfven}}$, this gives $\tau \sim U R_\star^4/G$.

A full time evolution equation can be obtained by summing the four individual components, and taking into account that the first two are only operational while the pair is within its birth cluster. This can then be used to infer the lifetime of a system with given initial separation. The fraction of binary stars within a critical separation $a_{\text{crit}}$ which will coalesce before a time $t$ is found to be $f_{\text{binary}}(a_{\text{crit}}) = \log(a_{\text{crit}}/a_{\text{min}})/\log(a_{\text{max}}/a_{\text{min}})$. Finally, the spectrum of system lifetimes can be integrated over in Equation (3), yielding the type Ia supernova rate. However, this exercise is unnecessary for an initial estimate of the supernova rate; because the dependence is logarithmic, this fraction only varies by a few percent over most of the interesting parameter range.

**Notes**

[1] Though our calculus explicitly counts the number of observers in the universe, throughout this paper we consider hypotheses for which conditions are clement for microbial life forms. Though these are certainly hardier than complex, macroscopic organisms, the presumption is that the abundance of the latter will track the former. We investigated various ways in which this assumption may be violated in a manner that impacts our calculations in [12], but so far have found few indications that the distinction makes a meaningful difference.

[2] A related issue is the measure problem, whereby if universes are taken to be spatially infinite, the expected number of observers per universe will also be infinite (see e.g., [21]). This introduces a problem because the comparison of number of observers per universe is then ill-defined, and highly sensitive to the method used to regulate these infinite values. Thankfully for our purposes,

however, these troubles mainly affect the cosmological parameters, and not the microphysical parameters we are concerned with here.

[3] Even within the Solar System this ratio varies from body to body, ranging from 0.7% for our Moon to 55% for Mercury. However, both of these are a result of giant impacts: typically, planetary composition will closely follow that of the host star in many respects [33].

[4] In principle, one could worry that for some parameter values the minimal mass black hole is smaller than the typical supernova remnant, precluding Ia supernovae. If we take $18 M_\odot$ as the smallest black hole creating progenitor [40], we find that white dwarfs will exist and comprise the majority of supernova remnants as long as $\alpha^2 < 31.3\beta$. This consideration only practically affects regions for parameter space where $\beta$ is 300 times smaller, and so can be safely disregarded.

[5] Application of the SEMF in this case actually predicts this reaction to be marginally allowed in our universe, a consequence of the actual value being below the SEMF's typical accuracy threshold. To correct for this, we set the additive multiple of the proton mass to correspond to the value we infer from the reverse reaction.

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
