# Peer review of "Multiverse Predictions for Habitability: Element Abundances"

_universe, doi:10.3390/universe8120651_

Round 1

Reviewer 1 Report

This is interesting and useful work. There is one caveat: the anthropic principle in the multiverse requires a solution to the problem of measure. This means that different measures lead to different responses. It would be interesting to see how the conclusions of the authors depend (and do they really!) on the choice of this or that measure. The topic is very important and has good potential for development. 

As for the improvement of methodology, to consider different cosmological measure to see how the conclusions of the authors depend (and do they really!) on the choice of this or that measure.

On the other hand, such a study may turn out to be non-trivial, and therefore I recommend this article for publication in the form in which it is presented.

Author Response

We thank the reviewer for prompting us to include a mention of the measure problem in our manuscript.  We now discuss it in footnote 2 on page 4.

While certainly important in any multiverse discussion, the measure problem mainly afflicts cosmological parameters, not the parameters we discuss, and so the problem can mainly be sidestepped.  We do plan to extend our analysis to cosmological parameters in a future work, and so will address the measure problem more thoroughly there.

Reviewer 2 Report

This is an interesting investigation on what habitability criteria should be picked in order to judge whether universes subject to different physical laws have a chance to emerge observers (the multiverse hypothesis). As far as I understand, the idea is to compute the probability to observe our values of a few fundamental constants - alpha, the electron to proton mass ratio, the gravitational strength in the field of a proton, and the ratios of up and down quark masses to the mass of the proton - within a habitability scenario and to dismiss or accept it based on whether these probabilities turn out to be much smaller or comparable to unity, respectively. In the present paper habitability is judged by elemental abundance ratios (metal to rock, organics to rock, C to O, etc.) whose values  are influenced by the abundances of SN Ia and SN II, alpha burning in stars as well as elemental stability (endpoints of nuclear decay series). Having determined a valid criterion one can use it to predict elemental abundance ratios and their associated values of fundamental constants which exclude the formation of complex life. 

There are a number of questionable assumptions in deriving the dependence of the various abundances on the fundamental constants, but these are made explicit and are discussed thoroughly by the authors. So I do not see a problem here. 

In general, the investigations performed in the present article are interesting and represents a new Ansatz for the discussion of variable fundamental constants in physics. 

Therefore, I recommend this work for publication in Universe. 

The authors may free the ms of three typos that I have spotted:

1) p.6.: by the

2) p.5: staple -> stable

3) p.14, Eq. (13) misses a point after the equation.

Author Response

We have fixed/clarified the typos the reviewer pointed out in the revision.